# Advances in Bidirectional Therapy for Peritoneal Metastases: A Systematic Review of Pressurized Intraperitoneal Aerosol Chemotherapy (PIPAC) Combined with Systemic Chemotherapy

**DOI:** 10.3390/cancers17152580

**Published:** 2025-08-06

**Authors:** Manuela Robella, Marco Vitturini, Andrea Di Giorgio, Matteo Aulicino, Martin Hubner, Emanuele Koumantakis, Felice Borghi, Paolo Catania, Armando Cinquegrana, Paola Berchialla

**Affiliations:** 1Unit of Surgical Oncology, Candiolo Cancer Institute, FPO-IRCCS, 10060 Candiolo, Italy; 2Centre for Biostatistics, Epidemiology and Public Health, Department of Clinical and Biological Sciences, University of Turin, 10043 Turin, Italy; marco.vitturini@edu.unito.it (M.V.); emanuele.koumantakis@unito.it (E.K.);; 3Surgical Unit of Peritoneum and Retroperitoneum Surgery, Fondazione Policlinico Universitario Agostino Gemelli IRCCS, 00168 Rome, Italy; 4General Surgery Department, Universita’ Cattolica del Sacro Cuore, 00168 Rome, Italy; 5Department of Visceral Surgery, Lausanne University Hospital, University of Lausanne, 1005 Lausanne, Switzerland; 6Department of Surgical Sciences, Catholic University of the Sacred Heart, 00168 Rome, Italy

**Keywords:** Pressurized Intraperitoneal Aerosol Chemotherapy (PIPAC), systemic chemotherapy, chemotherapy, bidirectional treatment, systematic review

## Abstract

Peritoneal metastases represent a challenging and often late-stage manifestation of several gastrointestinal and gynecological cancers. These patients frequently experience limited treatment options and poor outcomes. In recent years, a novel strategy combining intraperitoneal and systemic chemotherapy—also known as bidirectional treatment—has emerged to improve local and systemic disease control. Among these innovative approaches, Pressurized Intraperitoneal Aerosol Chemotherapy (PIPAC) allows for direct administration of chemotherapy into the abdominal cavity under pressure, aiming to increase drug penetration and reduce systemic toxicity. This systematic review evaluates the current clinical evidence on the combined use of PIPAC and systemic chemotherapy. The findings highlight promising survival outcomes, manageable toxicity, and the need for more standardized approaches. This work may support clinicians and researchers in refining treatment strategies and guiding future clinical trials for patients with peritoneal metastases.

## 1. Introduction

### 1.1. Background and Evolving Role of PIPAC

Peritoneal metastases (PM) represent a frequent and clinically challenging manifestation in various gastrointestinal and gynecological malignancies, typically associated with poor prognosis and limited treatment options. Patients with colorectal cancer and peritoneal metastases typically have a median overall survival of around 5 to 6 months, compared to around 12 to 16 months with modern systemic therapy [1]. In gastric cancer, the median survival for peritoneal metastases remains between 2 and 9 months even when treated with chemotherapy [2]. In ovarian cancer, the 5-year survival rate declines significantly with advanced stage, reaching approximately 25% in stage III and 15% in stage IV disease [3]; in pancreatic cancer, prognosis remains extremely poor, with a 5-year survival rate of only 8% following initial diagnosis [4].

The limited efficacy of systemic chemotherapy against PM—primarily due to the peritoneal–plasma barrier—has catalyzed the development of innovative locoregional therapies, notably Pressurized Intraperitoneal Aerosol Chemotherapy (PIPAC) [5].

Initially introduced in the palliative setting, PIPAC permits repeated laparoscopy-guided administration of aerosolized chemotherapy within a pressurized capnoperitoneum. This technique enhances drug penetration into peritoneal tissues while maintaining minimal systemic toxicity [6]. Early-phase studies have demonstrated objective tumor response rates exceeding 50% in selected cohorts [7,8,9,10].

Although originally developed for symptom control and tumor stabilization in non-resectable cases, the clinical use of PIPAC is rapidly evolving. Recent studies have explored its integration in neoadjuvant settings to downstage disease before cytoreductive surgery as well as adjuvant and even prophylactic strategies to prevent peritoneal dissemination in high-risk patients [11,12,13]. These findings suggest a shift in the perception of PIPAC from a purely palliative intervention towards a versatile therapeutic option across the disease spectrum.

A previous systematic review by Ploug et al. [14] emphasized the feasibility of bidirectional therapy but highlighted significant limitations, including predominantly retrospective data, small sample sizes, and inconsistent outcome reporting. Given recent advancements and emerging evidence from both retrospective and prospective trials, an updated comprehensive synthesis is now warranted.

Accordingly, this systematic review aims to rigorously assess overall survival (OS), progression-free survival (PFS), health-related quality of life (QoL), and treatment-related morbidity associated with bidirectional chemotherapy, thereby contributing robust insights to clinical decision-making.

### 1.2. Rationale for Bidirectional Chemotherapy

The rationale for combining PIPAC with systemic chemotherapy—commonly referred to as bidirectional treatment—is based on the complementary strengths of the two approaches. While systemic chemotherapy targets circulating tumor cells and distant metastases, it is often limited in the peritoneal compartment [15]. PIPAC, on the other hand, achieves high local drug concentrations in the peritoneum, but lacks systemic coverage. Therefore, their combination aims to maximize oncological control by targeting the disease from both local and systemic directions. Importantly, the concept of bidirectional treatment is not new. Its origins trace back to strategies combining systemic and intraperitoneal chemotherapy via HIPEC (Hyperthermic Intraperitoneal Chemotherapy) or catheter-based approaches, especially in ovarian and gastric cancers [16,17]. What differentiates PIPAC is its repetitive and minimally invasive nature, allowing for sequential treatments with ongoing monitoring of response and tolerability [13].

### 1.3. Operational Definition of Bidirectional Chemotherapy

Despite increasing clinical adoption, definitions of bidirectional treatment remain inconsistent across the literature. For clarity and consistency, this review operationally defines bidirectional treatment as: *the combination of PIPAC and systemic chemotherapy administered concurrently, sequentially between PIPAC cycles, or continuously throughout the treatment course*. Importantly, systemic chemotherapy administered only prior to the first or after the final PIPAC cycle does not meet this definition. Eligible routes of administration include intravenous, intraperitoneal, or other novel delivery methods.

We acknowledge the methodological challenges inherent in evaluating bidirectional therapy, including variability in chemotherapeutic regimens, procedural techniques, and outcome measurement. Therefore, this review critically assesses the presence of standardized protocols and systematic outcome reporting in the available literature.

## 2. Methods

### 2.1. Protocol and Study Registration

This review was registered with the international prospective register of systematic reviews (PROSPERO) in accordance with PRISMA-P guidelines (PROSPERO https://www.crd.york.ac.uk/PROSPERO/view/CRD420251025681, accessed on 31 July 2025, (CRD420251025681)) [18].

### 2.2. Literature Search

This systematic review—including the literature search, study selection, data extraction, and synthesis—was conducted in accordance with the Preferred Reporting Items for Systematic Reviews and Meta-Analyses (PRISMA) guidelines [19]. We conducted a systematic search of MEDLINE (via Ovid), Embase (via Ovid), the Cochrane Central Register of Controlled Trials (CENTRAL), Scopus, and ClinicalTrials.gov for articles published from January 2011 through April 2025. Search terms combined controlled vocabulary (e.g., PIPAC, “pressurized intraperitoneal aerosol chemotherapy”) and free-text keywords for peritoneal metastasis and systemic chemotherapy. Full database-specific search strategies are available in the systematic review registration (PROSPERO: https://www.crd.york.ac.uk/PROSPERO/view/CRD420251025681, accessed on 31 July 2025, (CRD420251025681)), specifically in the https://www.crd.york.ac.uk/PROSPEROFILES/70483cfa62e8a56671a24f1f72c3b3b7.pdf, accessed on 31 July 2025, “link to search strategy” section.

### 2.3. Study Selection

Two reviewers (M.V., P.B.) independently screened titles and abstracts for eligibility. Where conflicts emerged, a third author (M.R.) resolved the conflict. We included human clinical studies of adult patients (≥18 years) with peritoneal metastasis treated with pressurized intraperitoneal aerosol chemotherapy (PIPAC) combined with systemic chemotherapy. We excluded case reports, reviews, and studies without separate data on bidirectional treatment. Full texts of potentially eligible articles were assessed against predefined criteria; disagreements were resolved by consensus.

### 2.4. Data Extraction

Using a piloted form, two reviewers (M.V., P.B.) independently abstracted the following:Study characteristics: Author, year, country, design, sample size, proportion receiving bidirectional therapy.Patient demographics: Age, sex, primary tumor origin, prior chemotherapy.Intervention details: PIPAC drugs and dosages, systemic regimens, treatment intervals.Outcomes: Overall survival (OS), progression-free survival (PFS), pathological response (e.g., PRGS), radiological response (e.g., RECIST criteria), quality of life (e.g., EORTC QLQ-C30, EQ-5D), and adverse events (Clavien–Dindo, CTCAE).

We denoted each study’s country of origin using ISO 3166-1 alpha-3 country codes, which is the internationally standardized three-letter country code system [20]. When outcomes were reported using different summary statistics (e.g., median [range], mean [SD], median [IQR]), we retained each study’s original measures in order to maximize inclusion while avoiding data transformation and potential loss of information. Any discrepancies were reconciled by discussion.

### 2.5. Methodological Quality Assessment

To assess methodological quality and potential bias, we applied the Methodological Index for Non-Randomized Studies (MINORS), a validated instrument comprising twelve items for comparative and eight items for non-comparative studies [21]. Each study was scored 0–2 on domains such as inclusion of consecutive patients, unbiased endpoint assessment, follow-up adequacy, and statistical analyses adapted to study design.

Although no formal consensus on MINORS thresholds exists, we adopted cutoffs from recent oncology and broader medical systematic reviews [22,23]—≤ 8 denoting poor quality, 9–14 moderate quality, and 15–16 good quality—because these categories are well suited to appraising non-randomized observational designs.

### 2.6. Data Synthesis

Given the clinical and methodological heterogeneity, quantitative meta-analysis was not performed. We conducted a structured narrative synthesis, organizing results into five domains with corresponding tables:Study characteristics (design, demographics, prior treatment).PIPAC and systemic chemotherapy regimens.Survival and tumor response outcomes.Quality of life measures and longitudinal scores.Adverse event rates (Clavien–Dindo, CTCAE) and treatment discontinuation.

Where both bidirectional and monotherapy cohorts were reported, direct comparisons and subgroup differences by tumor type are highlighted. Each outcome is presented in its original statistical format to preserve fidelity and completeness.

## 3. Results

### 3.1. Study Selection

#### 3.1.1. Study Flow

The systematic search (January 2011–April 2025) retrieved 614 records from MEDLINE, 1397 from EMBASE, and 171 from CENTRAL plus 30 from ClinicalTrials.gov, as summarized in the PRISMA 2020 flow diagram (Figure 1) generated with the PRISMA2020 V 1.1.1 R software package [24].

After removal of 821 duplicates—reflecting robust indexing across the three databases—1391 unique records remained for title and abstract screening. We excluded 1275 records at this stage, leaving 116 full-text reports for detailed evaluation. Six reports could not be retrieved despite multiple attempts. Of the 109 full-text articles assessed, 83 were excluded for the following reasons: wrong study design (n = 9), wrong population (n = 17), wrong intervention (n = 16), wrong outcome (n = 16), wrong publication type (n = 22), and background articles (n = 3), resulting in 22 clinical studies and five additional reports included in this review. Citation tracking was not performed due to the large initial volume of retrieved records. The search was limited to January 2011 onwards, corresponding to the first in-human PIPAC application [5], and to English-language publications only, given its role as the primary medium for international scientific communication.

#### 3.1.2. Excluded Studies

As a result of the full text screening process, 83 papers were excluded (n = 83). The most frequent exclusion criteria were wrong publication type (n = 22) and wrong population (n = 17), followed by wrong outcome (n = 16) and wrong intervention (n = 16), while six (n = 6) potentially eligible reports could not be obtained for assessment.

### 3.2. Study Characteristics

A total of 22 clinical studies published between 2015 and 2024 were included in this systematic review. Of these, 15 (68%) were retrospective and 7 (32%) prospective.

The included studies were conducted across a wide geographical spectrum, comprising Europe (Italy, France, Germany, Denmark, the UK, Switzerland), North America (USA), and Asia (Singapore, Russia). Sample sizes varied considerably, ranging from small early-phase feasibility cohorts (e.g., 3–20 patients in Reymond et al. [25]) to larger observational series exceeding 100 participants (e.g., Kryh-Jensen et al. [26], Siebert et al. [27]). Across all studies, a total of 1010 patients were analyzed, with 742 (approximately 73%) receiving concurrent or sequential bidirectional therapy.

Details regarding study design, geographical distribution, duration, sample size, bidirectional therapy and sex distribution (reported here because the primary aim of each study (e.g., focusing on ovarian versus gastrointestinal malignancies) directly influences the male-to-female ratio) are provided in Table 1.

### 3.3. Patient Characteristics

Patients presented with a range of primary tumors, most commonly gastric cancer (GC), colorectal cancer (CRC), and ovarian cancer (OC). Other malignancies included appendiceal cancer (AC), diffuse malignant peritoneal mesothelioma (DMPM), pseudomyxoma peritonei (PMP), and pancreatic cancer (PC). The median age of patients ranged between 44 and 66 years across studies; in the vast majority of cohorts, a significant proportion of patients (100% in 11 studies) had received prior systemic chemotherapy before entering bidirectional protocols. These patient demographics and treatment histories are summarized in Table 1.

### 3.4. Treatment Regimens

Across the 22 included studies, bidirectional treatment strategies combined PIPAC with various systemic regimens. While the intraperitoneal component tended to follow standardized drug protocols, systemic chemotherapy and scheduling were more heterogeneous across cohorts. PIPAC protocols predominantly included either:Oxaliplatin (92 mg/m^2^), mainly for colorectal and appendiceal cancers; orCisplatin (7.5 mg/m^2^) combined with doxorubicin (1.5 mg/m^2^), commonly used in gastric, ovarian, and other peritoneal surface malignancies.

In one study, PIPAC treatment with mytomicin C was performed in six patients. Notably, Casella et al. [30] employed an escalated PIPAC dosage of cisplatin 10.5 mg/m^2^ and doxorubicin 2.1 mg/m^2^, referencing the safe upper limits previously established in a Phase I study by Tempfer et al. [7].

In addition, electrostatic precipitation (ePIPAC)—a technical variant designed to enhance aerosol deposition—was used in Reymond et al. [25] as well as in Van de Vlasakker et al. [29] and more extensively within a retrospective cohort by Kryh-Jensen et al. [26], where 33 out of 108 patients underwent ePIPAC.

Among all included studies, two were phase I trials specifically designed to assess safety and feasibility as primary endpoints. In Sundar et al. [28], the combination of PIPAC-oxaliplatin 92 mg/m^2^ with nivolumab 240 mg every two weeks was tested in patients with gastric cancer and peritoneal metastases. Their study did not employ dose escalation but rather aimed to establish the safety and tolerability of the chemo-immunotherapy pairing. Raoof et al. [32] investigated the safety of PIPAC-oxaliplatin 90 mg/m^2^ administered after systemic 5-fluorouracil (5-FU) and leucovorin (LV). The primary endpoint consisted of treatment-related adverse events.

Regarding systemic chemotherapy, regimens varied based on primary tumor origin, institutional preferences, and treatment intent (neoadjuvant vs. palliative). Commonly used systemic protocols included:FOLFOX, FOLFIRI, FOLFOXIRI, XELOX in gastrointestinal tumors.FLOT in advanced gastric cancer.Platinum-based doublets in ovarian and mesothelial malignancies.Occasional use of targeted therapies (e.g., bevacizumab, EGFR inhibitors) and immune checkpoint inhibitors.

The timing and integration of systemic chemotherapy relative to PIPAC were inconsistently reported. In most studies, systemic treatment was delivered continuously or in the intervals between PIPAC sessions.

The median interval between PIPAC procedures ranged from 3 to 8 weeks, with some protocols allowing variation based on patient response, toxicity, or logistics. Details about PIPAC and systemic treatment scheduling are reported in Table 2.

### 3.5. Survival and Tumor Response

Clinical outcomes varied substantially across the included studies, reflecting heterogeneity in patient selection, primary tumor types, treatment intent, and follow-up duration (Table 3).

The median follow-up, when reported, ranged from 2.4 months (Hilal et al. [41]) to 29.6 months (Kepenekian et al. [11]), although many studies did not specify a follow-up time or used non-standardized formats. This variability limits direct comparison of long-term outcomes.

Across the included studies, a total of 1637 PIPAC procedures were performed, with a median of 50 PIPACs per study and a median of 2.5 procedures per patient.

Among studies reporting survival outcomes, the median overall survival (OS) from the first PIPAC was 10.5 months, while the median OS from the time of diagnosis was 16.1 months. The most favorable outcomes were observed in gastric cancer cohorts. Casella et al. [30] reported a median OS of 10.5 months from the first PIPAC in patients with synchronous gastric peritoneal metastases, while Sundar et al. [28] observed a shorter OS of 6.0 months in a phase I study combining PIPAC-oxaliplatin with nivolumab. In colorectal cancer populations, Raoof et al. [32] reported a median OS of 12.0 months, while Kryh-Jensen et al. [26]. documented stratified outcomes with 16.0 months in colorectal, 10.0 months in pancreatic, and 7.8 months in gastric cancer, all calculated from the start of PIPAC treatment. In contrast, outcomes in biliary tract cancers were more limited, with Falkenstein et al. [37] reporting a median OS of just 2.8 months. Progression-free survival (PFS) was less consistently reported (in four studies), with a range from 1.8 to 12.0 months and a median of 4.5 months.

Radiological response (RR), based on Response Evaluation Criteria in Solid Tumors (RECIST) criteria, was reported in five studies, ranging from 7% to 71%, with a median response of 50%.

Histological tumor response was assessed in most of the included studies, primarily using the Peritoneal Regression Grading Score (PRGS), which assigns lower values to greater response (1 = complete response, 4 = no response). Two studies instead applied the Tumor Regression Grade (TRG) system, where the scale is inverted, i.e., higher scores indicate better tumor regression (0 = no response, 4 = complete response).

In gastric cancer patients treated with bidirectional therapy, Sundar et al. [28] reported PRGS grades 1–2 in 66.7% of cases after the second PIPAC and in 100% of patients who underwent three procedures, suggesting a cumulative histologic effect. Raoof et al. [32] documented PRGS reduction in 42% of patients. Consistently favorable results were also seen in other studies; Di Giorgio et al. [35] described PRGS 1–2 in 61.5%, and Casella et al. [30] reported PRGS 1–2 in 67% of patients at the second PIPAC and 4.8% at the third PIPAC. Three studies assessed histological response using the TRG system. Falkenstein et al. [37] observed major or complete regression (TRG 3–4) in 40% of patients with biliary tract cancers. Khosrawipour et al. [42] reported a comparable rate of TRG 3–4 in 35% of patients with gastric cancer. The highest rate was observed in the pilot study by Demtroeder et al. [43], in which 65.2% of patients showed a major or complete histological response.

Macroscopic disease burden was assessed through the Peritoneal Cancer Index (PCI) in several studies, although detailed follow-up data across treatment cycles were limited. Baseline PCI values were reported in most cohorts; however, only a few studies documented longitudinal changes over the course of PIPAC treatment. In Sundar et al. [28], the baseline median PCI was 20, with a reduction of 5 points observed after the second PIPAC and 7 points after the third; similarly, Raoof et al. [32] described a decrease in PCI in 50% of patients. In Kepenekian et al. [11], which was conducted in malignant peritoneal mesothelioma, the median PCI decreased modestly from 27 to 25. Alyami et al. [39] described a gradual and modest reduction in median PCI across successive PIPAC cycles, suggesting a trend toward disease control over time. Conversely, Di Giorgio et al. [35] reported a reduction in PCI in only 15.4% of patients, with stable disease in 7.7%.

The principal survival and tumor-response metrics discussed above are summarized in Table 3 while clinical outcomes, stratified for underlying pathology, were summarized in Table 4.

Data on peritoneal cytology were more limited and inconsistently reported. In Sundar et al. [28], baseline positivity was noted in 77.8% of cases, although post-treatment cytology was not documented. Kryh-Jensen et al. [26] provided cytological conversion data in a subset of patients, indicating rare clearance among long-term survivors (14%). Interestingly, despite the limited macroscopic response, a more nuanced picture emerges through the Combined Progression Index (CPI), a composite surrogate integrating PRGS and cytology dynamics across PIPAC cycles. In this cohort, CPI 2—indicative of disease control—was observed in 69% of patients, while CPI 3—suggesting progression—was noted in only 8%.

Ascites was inconsistently reported across the included studies. While it is a clinically relevant parameter in patients with peritoneal metastases, most studies either did not mention ascites or only reported its presence at baseline without follow-up data. Among the included cohorts, only three studies reported changes in ascitic volume during the course of PIPAC treatment. Casella et al. [30] documented median ascitic volumes at each PIPAC: 499 mL (range 0–6800) at the first and 999 mL (range 0–7800) at the second, alongside a concurrent reduction in median PCI. Di Giorgio et al. [35] noted the presence of ascites in 10 out of 26 patients at baseline, with a median volume of 2750 mL (range 800–5000); ascites increased in four patients, decreased in three, and remained stable in the others. Alyami et al. [33] reported ascites in eight patients, with a baseline median volume of 1 L (range 500–4000 mL); after treatment, complete disappearance was observed in 50% of cases.

Quality of life (QoL) was explicitly evaluated in six (n = 6) studies, although with varying levels of detail and methodology. Validated tools such as EORTC QLQ-C30 and SF-36 were used in most of these, demonstrating either stable or improved QoL outcomes during PIPAC treatment. In the CRC-PIPAC-II study [29], improvements were noted in global health status as well as in emotional, cognitive, and role functioning. Similarly, the PIANO trial [28] and UK-PIPAC Trial [31] each reported no significant deterioration in QoL. QoL-PIPAC-PC-QoL, the only study reporting serial scores, documented minimal decline across the first two PIPAC procedures (baseline 66 ± 2.6; 64 ± 3.75 after the first PIPAC; 63 ± 3.85 after the second PIPAC). Although Kryh-Jensen et al. [26] confirmed the use of EORTC QLQ-C30, specific outcomes were not detailed. Lastly, Robella et al. [45] also reported preservation of QoL throughout treatment.

### 3.6. Safety and Treatment-Related Toxicity

Tolerability of bidirectional therapy combining PIPAC with systemic chemotherapy was variably reported across studies. Despite some heterogeneity in grading and documentation, the overall safety profile was acceptable, with most adverse events being mild to moderate.

High-grade complications were observed in specific clinical contexts, particularly where immunotherapy or advanced disease burden were involved. Surgical complications were explicitly reported in seven studies; overall morbidity rates were low, and most adverse events were minor. Four studies reported no surgical complications (0%) [29,36,43,45]. In the other three studies, major surgical morbidity (Clavien–Dindo ≥ 3) was 0–6% [30,34,35]. Medical complications graded by CTCAE were reported in 15 studies. In the majority of cases, toxicities were grade 1–2, such as abdominal discomfort, nausea, and fatigue, and were generally self-limiting. However, grade ≥ 3 events were observed in multiple studies, with the most notable rates in early-phase or combination protocols. Grade 3 toxicity events ranged from 2% to over 50%, while grade 4 events were less frequent but still documented in selected cohorts. Fatal complications (grade 5) occurred in a few studies, with reported rates between 2.4% and 11.1%, often linked to bowel perforation or sepsis.

Treatment discontinuation was explicitly reported in a limited number of studies, with variable frequencies and causes. The most commonly cited reason was disease progression, followed by clinical deterioration, adverse events, and technical challenges such as non-access due to peritoneal adhesions. Feldbrügge et al. [34] reported the highest discontinuation rate, with 74% of patients stopping treatment, largely due to progression, patient or oncologist decision, and access difficulties. Similarly, Demtroeder et al. [43] documented a 35% discontinuation rate, attributing it to adhesions or disease evolution. In Robella et al. [45], the dropout rate was lower (6%) and exclusively due to intraoperative adhesions. Raoof et al. [32] indicated that half of the patients discontinued PIPAC, all due to radiological or clinical progression. Although Kyle et al. [31] and Khomyakov et al. [44] did not specify percentages or number of discontinuations, they both noted treatment interruption due to either progression or toxicity. Across the remaining studies, reasons for discontinuation were either not specified or entirely absent, limiting the ability to fully evaluate treatment feasibility and adherence.

An overview of surgical and medical complications together with other key adverse events reported across the included studies is presented in Table 5.

### 3.7. Forthcoming Bidirectional PIPAC Trials

Several prospective trials are currently investigating the addition of PIPAC to standard systemic therapies in gastric, pancreatic, and colorectal peritoneal metastases (Table 6).

The PIPOX02 randomized phase II study (Dumont et al. [46], 2025) in France will compare systemic FOLFIRI/FOLFIRINOX with or without PIPAC oxaliplatin (up to four administrations) in colorectal peritoneal metastases. Its primary endpoint is PFS, with OS, ORR, safety, QoL, PK, and peritoneal fluid/tumor biomarker analyses as secondary measures.

For pancreatic cancer, the Nab-PIPAC trial (Di Giorgio et al. [47], 2024; NCT05371223) is a monocentric open-label phase II study of systemic nab-paclitaxel/gemcitabine plus intraperitoneal nab-paclitaxel delivered via PIPAC. It is powered for disease-control rate (RECIST v1.1) and will also assess safety, pathological response (PRGS), PFS, OS, QoL, pharmacokinetics, and translational biomarkers.

In Lithuania, the Lukšta phase II trial (Lukšta et al. [48], 2023; NCT05644249) will evaluate first-line PIPAC (cisplatin + doxorubicin) combined with six cycles of FOLFOX in previously untreated gastric cancer with peritoneal metastases. Its primary objective is ORR (RECIST v1.1) after four cycles of FOLFOX, with secondary endpoints including procedural feasibility, PCI/PRGS kinetics, safety, QoL, PFS, and OS.

The PIPAC VEROne study (Casella et al. [49], 2022; NCT05303714) is a multicentric open-label phase III randomized trial in Italy that will compare FOLFOX alone versus FOLFOX plus PIPAC (cisplatin + doxorubicin) in oligometastatic gastric cancer (PCI ≤ 6), with R0 resectability as the primary endpoint and OS, PFS, histologic response (PRGS/TRG), and QoL among key secondary outcomes.

**Table 6 cancers-17-02580-t006:** Summary of selected reports’ provenance, study design, intervention, prospective number of participants, primary and secondary outcomes, current status, and estimated completion date.

Trial (Author & Year)	Country *	Design	Tumor/Eligibility	Intervention	N	Outcomes	Status	Est. Completion
Dumont et al., 2025 [46]	FRA	PRO; II randomized; multicentric	CRC/PCI > 15; ECOG 0–2	FOLFIRI/FIRINOX ± PIPAC Oxaliplatin	114	Prim: PFS; Sec: OS, ORR, safety, QoL, PK, biomarkers	Not yet recruiting	August 2029 ^§^
Di Giorgio et al., 2024 [47]	ITA	PRO; II; Single-arm; open label; monocentric	PC-PM; ECOG 0–1	IV Nab-Pac Gem + PIPAC Nab-Pac	38	Prim: RR (RECIST); Sec: safety (CTCAE, CD), PRGS, PFS, OS, QoL, PK, biomarkers	Recruiting	July 2025 ^†^
Casella et al., 2022 [49]	ITA	PRO; III randomized; open label; multicentric	Oligometastatic GC-PM; PCI ≤ 6; ECOG 0–1	FOLFOX ± PIPAC (Cis + Dox)	98	Prim: R0 resectability; Sec: OS, PFS, histology (PRGS/TRG), QoL, AEs, cost-effectiveness	Recruiting	April 2028 ^‡^
Luksta et al., 2023 [48]	LTU	PRO; II; Single-arm, open label; multicentric	GC-PM; ECOG 0–1; 1L FOLFOX	3 × PIPAC (Cis + Dox) + 6 × FOLFOX	37	Prim: RR (RECIST); Sec: procedural metrics, PCI/PRGS, ascites, safety, QoL, PFS, OS	Recruiting	—
Sgarbura et al., 2019 [50,51]	FRA	PRO; II; randomized; open label; multicentric	Mid/low RC; ≥2 MRI-risk; ECOG 0–2	Cis + Pem ± PIPAC (Cis + Dox)	66	Prim: OS; Sec: PFS, safety, compliance, feasibility, conversion to resectability, histological response, QoL	Completed	December 2024
Eveno et al., 2018 [52]	FRA	PRO; II; randomized; open label; multicentric	GC-PM; PCI > 8; WHO 0–1	PIPAC (Cis + Dox) ± IV chemo	94	Prim: PFS; Sec: OS, safety, QoL, morbidity, secondary resectability	NR	—

Abbreviations: Locoregional therapy—PIPAC = Pressurized Intraperitoneal Aerosol Chemotherapy. Tumor/eligibility—CRC = colorectal cancer; GC-PM = gastric cancer with peritoneal metastasis; PC-PM = pancreatic cancer with peritoneal metastasis; RC = rectal cancer; PCI = Peritoneal Cancer Index; ECOG = Eastern Cooperative Oncology Group performance status; 1L = first-line therapy; MRI = magnetic resonance imaging; WHO = World Health Organization performance status. Outcomes—PFS = progression-free survival; OS = overall survival; ORR = objective response rate; RR = response rate; QoL = quality of life; PK = pharmacokinetics; PRGS = Peritoneal Regression Grading Score; TRG = Tumor Regression Grade; AEs = adverse events; R0 = microscopically margin-negative resection. Methodology/assessment—RECIST = Response Evaluation Criteria in Solid Tumors; CTCAE = Common Terminology Criteria for Adverse Events; CD = Clavien–Dindo classification. Study design—PRO = prospective study; RCT = randomized controlled trial; Ph II/III = phase II/III. * ISO 3166-1 alpha3 country codes. ^†^ ClinicalTrials.gov primary completion 30 July 2025. ^‡^ Protocol: enrollment + 36 mo follow-up. ^§^ ClinicalTrials.gov primary completion August 2029.

The MESOTIP trial (Sgarbura et al. [50,51], 2019) is a completed multicentric phase II RCT that randomized 66 patients with diffuse malignant peritoneal mesothelioma to cisplatin–pemetrexed with or without adjunctive PIPAC (cisplatin + doxorubicin). Overall survival is the primary endpoint, while PFS, safety, feasibility, conversion to resectability, histological response, and QoL are among the prespecified secondary outcomes.

Finally, the French *EstoK 01* study (Eveno et al. [52], 2018) is an open-label phase II trial planning to enroll 94 patients with gastric peritoneal metastases to receive systemic chemotherapy with or without bidirectional PIPAC (cisplatin + doxorubicin). Progression-free survival is the primary outcome, while OS, safety, QoL, morbidity, and potential for secondary resection constitute key secondary endpoints. The trial is not listed on ClinicalTrials.gov; it was retrieved separately for relevance from the published protocol, and according to that report had not yet begun recruitment during the year 2018. Although final results have not yet been published, preliminary data have been presented at international conferences. The trial was discontinued due to toxicity, possibly related to suboptimal patient selection.

Collectively, these trials—spanning not-yet-recruiting, actively recruiting and completed phases—will clarify the clinical utility of bidirectional therapy across tumor types and inform optimal patient selection, procedural protocols, and outcome assessment. However, because none of these studies directly compare bidirectional PIPAC-based approaches with systemic therapy alone, the relative efficacy of adding intraperitoneal chemotherapy remains to be established, underscoring the critical need for trials to fill that specific evidence gap.

### 3.8. Methodological Quality Assessment

Overall methodological quality was moderate; among the 21 non-randomized studies, the median MINORS score was 13 of 16 (81%), with individual scores ranging from 11 to 14—all falling within the predefined “moderate” bracket (n = 21, 100%) as shown in Table 7 and Table 8. Items I1–I4 (clear study aim, consecutive patient inclusion, prospective data collection, and appropriate endpoints) uniformly achieved the maximum score, reflecting strong adherence to foundational design principles. A notable shortfall was seen for item I5 (blinding of outcome assessment), which rarely scored full marks—while blinding in procedural cohorts can be inherently difficult, authors should explicitly state when and why it is not feasible in order to satisfy this MINORS criterion and mitigate potential detection bias. Item I8 (prospective sample size calculation) was the least addressed, receiving zero points in the majority of reports, likely because formal power analyses are not routinely conducted in early observational cohorts. Moreover, even when sample size was reported, the rationale often lacked a formal power calculation and instead reflected expected cohort size; preregistered power analyses would strengthen methodological rigor. Item I7 (loss to follow-up ≤ 5%) also showed notable variability, suggesting challenges in complete patient tracking, often due to the palliative setting of the treatment.

While many of the included non-comparative cohorts adhered to key methodological standards, the inherent limitations of such designs—particularly regarding a priori sample size determination and follow-up completeness—may introduce bias that undermines the precision and internal validity of reported outcomes. Notably, Siebert et al., 2021 [27] stands apart as the only comparative cohort (MINORS I9–I12), achieving balanced baseline characteristics and contemporaneous controls, which enhances its internal validity and allows for more robust inference of PIPAC effects.

Therefore, caution is warranted when interpreting findings from these primarily non-comparative studies, and more robust comparative designs should be pursued where feasible in future bidirectional PIPAC research. We recommend that observational investigations pre-register detailed protocols with explicit sample-size justifications, conform core variables, implement standardized follow-up procedures, and ensure blinded endpoint adjudication in order to strengthen comparability and reduce bias. We further encourage that future observational studies incorporate a comparative arm—ideally with contemporaneous controls and pre-specified matching or randomization—to strengthen causal inference and mitigate selection bias.

## 4. Discussion

This systematic review has examined the efficacy and safety of bidirectional treatment strategies combining PIPAC with systemic chemotherapy in patients with peritoneal metastases.

Building upon the foundational review by Ploug et al. [14] we analyzed 22 clinical studies published between 2015 and 2024 involving 1010 patients, 742 of whom received bidirectional treatment.

Early studies such as those by Robella et al. [45] and Larbre et al. [38] primarily assessed feasibility and safety, showing that PIPAC can be administered repeatedly with low procedural risk. Its use has expanded since then to include neoadjuvant and conversional strategies, and in selected cases adjuvant or prophylactic intent.

Median OS from the first PIPAC ranged from 2.8 to 19.1 months, with the best outcomes seen in the gastric cancer, colorectal cancer, and mesothelioma cohorts [11,26,30,39]. PFS was reported in fewer studies, and ranged from 1.8 to 12 months.

Histologic response—assessed via PRGS or TRG—was observed in over 60% of patients in most studies, and radiological response rates ranged from 7% to 66%. PCI reductions were noted in several cohorts, reinforcing the utility of laparoscopic restaging. While cytologic conversion was infrequent, composite indices such as the Combined Progression Index (CPI) may offer more integrated response measures.

Although cytological conversion from positive to negative was infrequent and appeared to have limited prognostic value, some studies—such as Kepenekian et al. [11]—introduced composite indices such as the Combined Progression Index (CPI), integrating cytology and PRGS as surrogate indicators of therapeutic response.

Treatment-related toxicity was generally low, with grade 1–2 adverse events being the most common, including abdominal discomfort, nausea, fatigue, and mild laboratory abnormalities. Grade ≥ 3 toxicities were reported in 15 studies, though these were not consistently associated with the use of biological agents or immunotherapy.

In particular, the combinations of PIPAC with bevacizumab (Siebert et al. [27], 2021) and with ramucirumab (Feldbrugge et al. [34], 2021) were reported to be safe, with no increase in severe toxicity or mortality. In the PIANO Study (Sundar et al. [28], 2024), all grade 5 events occurred in the non-bevacizumab arm and were reviewed by an independent data monitoring committee, which attributed them to disease progression rather than treatment. These findings support the feasibility of integrating biological agents into bidirectional protocols without compromising safety. Notably, hematologic toxicity such as neutropenia and thrombocytopenia was reported in multiple studies, and is likely attributable to the systemic chemotherapy component rather than to PIPAC itself. However, most studies failed to clearly distinguish whether adverse events were reported per patient or per procedure as well as whether they were related to the intraperitoneal or systemic route, highlighting a major limitation in toxicity attribution.

Surgical complications, assessed using the Clavien–Dindo classification, were infrequent and mostly minor (grade I–II). Grade ≥ 3 surgical events were reported only in a few studies (Di Giorgio et al. [35], Feldbrügge et al. [34]), with no related mortality, confirming the procedural safety of PIPAC in experienced centers.

Another important and unresolved issue concerns the optimal timing and sequencing between systemic chemotherapy and PIPAC, which varied substantially across studies. Some protocols adopted alternating cycles every 4–6 weeks, while others administered systemic therapy concurrently or continuously. Dedicated studies are needed to evaluate how interval length and treatment synchronization impact efficacy, tolerability, and patient selection.

Quality of life (QoL) is particularly relevant in the palliative context, and was assessed in six studies. Most of these employed validated instruments (EORTC QLQ-C30, SF-36) and reported stable or improved QoL scores. Although these findings are encouraging, more systematic and longitudinal evaluations of patient-reported outcomes are warranted.

In selected cases, bidirectional therapy enabled conversion to secondary CRS + HIPEC, supporting its potential not only for disease control but also as a bridge to curative surgery. Kepenekian et al. [11] reported conversion in 4 out of 32 (12.5%) patients with peritoneal mesothelioma, while Alyami et al. [33] documented 6 conversions among 26 (23%) patients. Though limited in scale, these findings underscore the potential of bidirectional therapy to downstage disease in selected individuals.

Despite encouraging results, several limitations persist. The methodological quality of all included studies was moderate, with no randomized trials available to date. Most studies were retrospective and characterized by heterogeneous systemic regimens, variable treatment intervals, and outcome definitions. Adverse event reporting was also inconsistent, and toxicity attribution between PIPAC and systemic components remains poorly defined. In many cases, it is unclear whether toxicities were reported per patient or per procedure, further complicating cross-study comparisons.

Furthermore, causes for treatment discontinuation were under-reported. When specified, progression of disease, development of adhesions, or clinical deterioration were among the most common causes, as noted in a few studies [34,43,45].

Taken together, these data support the feasibility, safety, and potential efficacy of bidirectional therapy in selected patients with peritoneal metastases, particularly from gastric and colorectal origins.

However, its optimal integration within standard treatment pathways—including timing, sequencing, and appropriate patient selection—remains to be clearly established.

To advance the field, future studies should not only focus on oncologic efficacy but also adopt standardized endpoints. These should include uniform histologic response grading, harmonized definitions for complications, validated quality-of-life instruments, and detailed reporting of treatment adherence. These improvements are essential to enhancing data comparability and improving clinical interpretability across future trials.

Several prospective trials are currently underway to address these gaps. Phase II and III studies in gastric, colorectal, and pancreatic cancers as well as peritoneal mesothelioma are investigating combinations of PIPAC with or without systemic chemotherapy. Their results will be key to refining patient selection, optimizing treatment protocols, and clarifying the role of bidirectional therapy within the landscape of peritoneal surface malignancies [46,47,48,49,50,52].

## 5. Conclusions

Bidirectional treatment combining PIPAC with systemic chemotherapy is emerging as a feasible and increasingly adopted strategy for selected patients with peritoneal metastases.

This approach has shown promising signals of histologic and macroscopic response across diverse tumor types along with an acceptable safety profile and in some cases the potential to downstage disease and enable secondary cytoreduction.

However, methodological heterogeneity, limited comparative data, and inconsistent reporting of key outcomes such as toxicity attribution, quality of life, and treatment adherence still limit definitive conclusions.

Ongoing randomized and prospective trials will be essential to clarifying the therapeutic value of bidirectional strategies and guiding their integration into standard oncologic care.

## Figures and Tables

**Figure 1 cancers-17-02580-f001:**
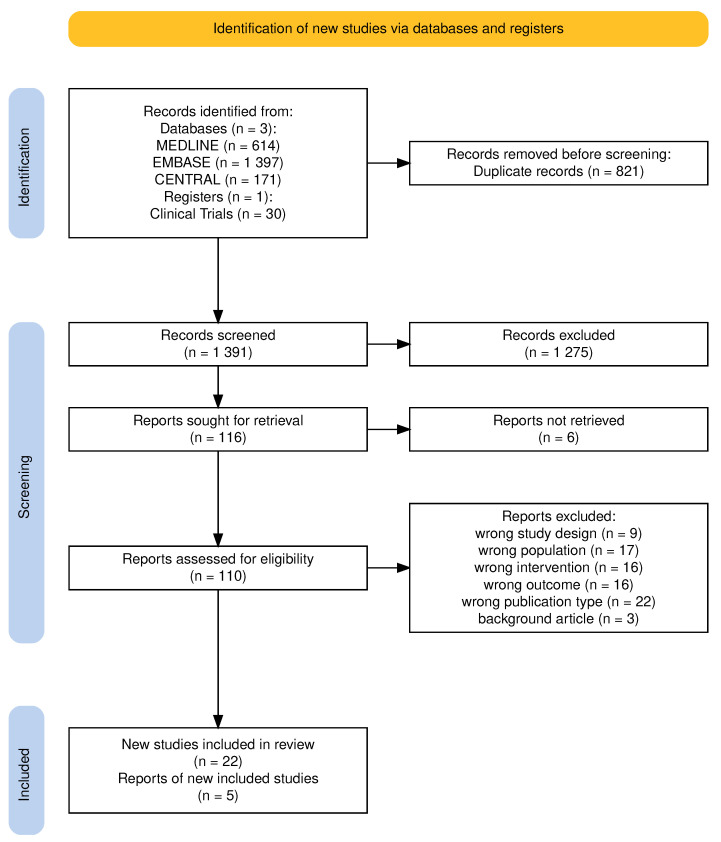
PRISMA flowchart of study selection.

**Table 1 cancers-17-02580-t001:** Study design, demographics, age, malignancies and prior chemotherapy.

Author & Year	Country *	Design **	Duration ***	Patients n	Bidirectional n (%)	Sex M (%)	Age ^†^	Malignancies ^‡^	Prev. Chemo n (%)
Sundar et al., 2024 [28]	SGP & BEL	PRO ^I^	Jun 20–Nov 22	18	18 (100%)	7 (39%)	65 ^a^ [52.5–70.75]	GC	18 (100%)
van de Vlasakker et al., 2024 [29]	NLD	PRO ^II^	Feb 20–Aug 22	20	20 (100%)	10 (50%)	57.5 ^a^ (41–70)	CRC, AC	0 (0%)
Casella et al., 2023 [30]	ITA	RETRO	Oct 19–Apr 22	42	42 (100%)	15 (36%)	60.5 ^a^ (49–68)	GC	NR
Kryh-Jensen et al., 2023 [26]	DNK	RETRO	Jan 16–Feb 23	108	49 (45%)	53 (49%)	GC 64 ^a^; PC 61 ^a^; CRC 69 ^a^	GC, PC, CRC	107 (99%)
Kyle et al., 2023 [31]	GBR	PRO	Jan 19–Jan 22	5	5 (100%)	2 (40%)	44 ^a^ (34–66)	CRC	5 (100%)
Raoof et al., 2023 [32]	USA	PRO ^I^	Aug 20–Jan 22	12	12 (100%)	7 (58%)	60 ^a^ (29–75)	AC, CRC	12 (100%)
Alyami et al., 2021 [33]	FRA	RETRO	NR–Jan 18	42	42 (100%)	20 (48%)	51.5 ^a^ (32–74.7)	GC	42 (100%)
Feldbruegge et al., 2021 [34]	DEU	RETRO	Mar 17–May 20	50	50 (100%)	28 (56%)	58 ^a^ (31–76)	GC	50 (100%)
Kepenekian et al., 2021 [11]	FRA	RETRO	Jan 16–May 20	26	26 (100%)	14 (54%)	64 ^a^ [56–69]	DMPM	9 (35%)
Siebert et al., 2021 [27]	FRA	RETRO	Dec 15–Mar 18	134	134 (100%)	60 (45%)	59.3 ^a^ (25–78)	GC, CRC, OC	134 (100%)
Di Giorgio et al., 2020 [35]	ITA	RETRO	Sep 17–Sep 19	28	26 (93%)	12 (43%)	50 ^a^ (38–79)	GC	28 (100%)
Gockel et al., 2020 [36]	DEU	PRO	Nov 15–Feb 18	13	13 (100%)	6 (46%)	61 ^a^ (49–77)	CRC, AC, small bowel	12 (66%)
Falkenstein et al., 2018 [37]	DEU	RETRO	Feb 13–Jan 17	13	3 (23%)	8 (61%)	58 ^b^ [37–75]	HPB	7 (54%)
Larbre et al., 2018 [38]	FRA	RETRO	Dec 15–Sep 17	43	39 (91%)	19 (44%)	59.8 ^b^ (33–77.9)	GC, OC, DMPM, PMP	39 (91%)
Alyami et al., 2017 [39]	FRA	RETRO	Dec 15–Dec 16	73	64 (88%)	31 (42%)	57.1 ^a^ (32.3–77.9)	GC, CRC, OC, DMPM, PMP	64 (88%)
Farinha et al., 2017 [40]	CHE	RETRO	Jan 15–Apr 16	42	1 (2%)	8 (19%)	66 ^a^ (59–73)	GYN, DIG	NR
Hilal et al., 2017 [41]	DEU	PRO	Apr 14–May 16	84	6 (7%)	0 (0%)	60.4 ^b^ ± 12.2	GYN	84 (100%)
Khosrawipour et al., 2017 [42]	DEU	RETRO	Jun 15–Jun 17	20	6 (30%)	15 (75%)	60.6 ^b^ (49–69)	PC	20 (100%)
Demtroeder et al., 2016 [43]	DEU	RETRO	Oct 12–Feb 14	17	11 (65%)	10 (59%)	59 ^b^ ± 12	CRC	16 (94%)
Khomyakov et al., 2016 [44]	RUS	PRO ^II^	Aug 13–Jul 16	31	31 (100%)	9 (29%)	52 ^b^	GC	7 (23%)
Reymond et al., 2016 [25]	DEU	RETRO	Jul 14–Oct 15	3	1 (33%)	1 (33%)	66.3 ^b^ (59–72)	PC, GBC	3 (100%)
Robella et al., 2016 [45]	ITA	RETRO	Jun 15–Feb 16	14	13 (93%)	NR	(18–78)	GC, CRC, OC, DMPM, PMP	14 (100%)
**Total [Mode] **	[DEU]	[R: 15 (68%)]	—	1010	742 (73%)	314 M: 453 F	—	[GC]	671 (66%)

* ISO 3166-1 alpha3 reported for each country. ** RETRO = “Retrospective”; PRO = “Prospective”; ^I^ Phase 1, ^II^ Phase 2. *** Dates shown as Mon YY (e.g., Feb 20 = February 2020). ^†^ Age values shown as median ^a^ [IQR] (range) or mean ^b^ ± SD. ^‡^ GC = Gastric Cancer; CRC = Colorectal Cancer; AC = Appendiceal Cancer; PC = Pancreatic Cancer; DMPM = Diffuse Malignant Peritoneal Mesothelioma; PMP = Pseudomyxoma Peritonei; OC = Ovarian Cancer; HPB = Hepato-pancreato-biliary; GBC = Gallbladder Cancer; GYN = Gynecological Cancer.

**Table 2 cancers-17-02580-t002:** PIPAC drug (dose), systemic chemotherapy regimens, interval between cycles, median PIPAC number for each patient, and overall number of PIPAC procedures performed.

Author & Year	PIPAC Drug (Dose mg/m^2^)	Systemic Regimen *	Median Interval Between PIPACs	Median PIPAC n **	Total PIPAC n
Sundar et al., 2024 [28]	OXA (90)	Nivolumab	6 wks	1 [1–3]	32
van de Vlasakker et al., 2024 [29]	OXA (92)	CapOX/FOLFOX/ FOLFIRI/ FOLFOXIRI + BEVA	1–4 wks post SC	NR	52; NR ^e^
Casella et al., 2023 [30]	CDDP + DXR (7.5/1.5 → 10.5/2.1)	FLOT/FOLFOX	6 wks	NR	74
Kryh-Jensen et al., 2023 [26]	OXA (NR)	NR	4–7 wks	3 (1–9)	NR; NR ^e^
Kyle et al., 2023 [31]	OXA (92)	NR	8 wks	2 (1–4)	10
Raoof et al., 2023 [32]	OXA (90)	FU + LV	6 wks	2 (1–3)	26
Alyami et al., 2021 [33]	CDDP + DXR (7.5/1.5)	XELOX	6 wks	3 (1–12)	163
Feldbrügge et al., 2021 [34]	CDDP + DXR (7.5/1.5)	NR ± Ramucirumab	20 d (7–41)	NR	90
Kepenekian et al., 2021 [11]	CDDP + DXR (7.5/1.5)	CDDP + MTA/GEM	6–8 wks	3 [1–3] (1–15)	79
Siebert et al., 2021 [27]	CDDP + DXR (7.5/1.5)/OXA (92)	NR ± BEVA	6 wks	3	397
Di Giorgio et al., 2020 [35]	CDDP + DXR (1.5/7.5)	ECF/mFOLFOX/5-FU/FOLFIRI/PTX	6–8 wks (2 wks post-SC)	NR	46
Gockel et al., 2020 [36]	CDDP + DXR (7.5/1.5)	FOLFOX + BEVA/ antiEGFR/FOLFIRI/ Cap+antiEGFR	6 wks	2 (1–6)	26
Falkenstein et al., 2018 [37]	CDDP + DXR (7.5/1.5)	NR	6 wks	NR	17
Larbre et al., 2018 [38]	CDDP + DXR (7.5/1.5)	NR	6 wks	3 (3–9)	175
Alyami et al., 2017 [39]	CDDP + DXR (7.5/1.5) OXA (92)/MMC (1.5)	NR	6–8 wks	2 (1–6)	164
Farinha et al., 2017 [40]	CDDP + DXR (7.5/1.5)	NR	6 wks	2 (1–4)	91
Hilal et al., 2017 [41]	CDDP + DXR (7.5/1.5)	NR	4–6 wks	NR	NR
Khosrawipour et al., 2017 [42]	CDDP + DXR (7.5/1.5)	GEM + Nab-PTX/FOLFIRINOX/ GEM	6 wks	NR	41
Demtröder et al., 2016 [43]	OXA (92)	NR	6 wks	3 (1–6)	48
Khomyakov et al., 2016 [44]	CDDP/DXR (7.5/1.5)	XELOX	6 wks	NR	56
Reymond et al., 2016 [25]	CDDP/DXR (7.5/1.5)	CDDP + GEM	6 wks	2 (2–6)	10; 4 ^e^
Robella et al., 2016 [45]	CDDP + DXR (7.5/1.5)/OXA (92)	Various SC	6 wks	3 (2–4)	40

* OXA = oxaliplatin; CDDP = cisplatin; DXR = doxorubicin; MMC = mitomycin C; 5FU = 5-fluorouracil; LV = leucovorin; CapOX = capecitabine + oxaliplatin; ECF = epirubicin + cisplatin + 5-FU; FOLFOX = LV + 5-FU+Oxaliplatin; mFOLFOX = modified FOLFOX; FOLFIRI = folinic acid + 5-FU + irinotecan; FOLFOXIRI = folinic acid + 5-FU + oxaliplatin + irinotecan; FLOT = 5-FU + LV + oxaliplatin + docetaxel; MTA = pemetrexed; Gem = gemcitabine; PTX = paclitaxel; Nab-PTX = nab-paclitaxel; BEVA = Bevacizumab; wks = weeks; d = days. ** Median number of PIPAC procedures administred reported as: Median [IQR] or Median (Range). ^e^ Total number of e-PIPAC.

**Table 3 cancers-17-02580-t003:** Clinical outcomes: follow-up, survival, radiological, clinical and pathological response.

Author & Year	Follow-Up (mo)	OS (mo)	PFS (mo)	RR *	PRGS	PCI
Sundar et al., 2024 [28]	20.8	6.0 (2.4–18.4) ^p^	1.8 (1.6–6.0) ^p^	7%; SD 57%	1–2 in 66.7% _2_; 100% _3_	20 [12.3–30]; Δ −5/−7 at _2,3_
van de Vlasakker et al., 2024 [29]	NR	NR	NR	NR	NR	27 (15–38)–NR
Casella et al., 2023 [30]	NR	19.6 (14–24) ^d^; 10.5 (7–13) ^p^	NR	NR	1: 37.1%; 2: 31.0%; 3: 21.4% _2_; 1: 2.4%; 2: 2.4%; 3: 9.5% _3_	16 [8–26]–NR
Kryh-Jensen et al., 2023 [26]	NR	GC 7.8; PC 10.0; CRC 16.0 ^p^	NR	NR	GC ↓ 61%; PC ↓ 59%; CRC ↓ 69%	NR
Kyle et al., 2023 [31]	11.6	11.6 (4.7–3.6) ^p^; 25.0 (7.2–42.2) ^d^	6.0 (0.5–25.1) ^p^	NR	NR	25 (20–32)–NR
Raoof et al., 2023 [32]	5.4	12.0 [4.3–NR] ^p^	2.9 [1.4–6.6] ^p^	SD 50%; PD 50%	↓42%; SD 17%; ↑ 42%	28 [19–32]; ↓ 50%; ↑ 50%
Alyami et al., 2021 [33]	NR	19.1 ^p^	NR	NR	NR	17 (1–39)–NR
Feldbrügge et al., 2021	NR	NR	NR	NR	NR	19 (1–39)–NR
Kepenekian et al., 2021 [11]	29.6	12.0	12.0	15%	median pre 3.0 (2.2–4.0); median post 2.0 (2.0–3.0)	pre 27 (20–34); post 25 (20–39)
Siebert et al., 2021 [27]	NR	NR	NR	NR	NR	18 (0–39)–NR
Di Giorgio et al., 2020 [35]	NR	12.3 (11.7–17.4) ^d^	NR	NR	CR (1) 7.7%; PR (2–3) 53.8%; SD (4) 38.5%	20 ± 9.9; ↑ 76.9%; SD 7.7%; ↓ 15.4%
Gockel et al., 2020 [36]	NR	10.1 (1–16.3) ^p^	NR	NR	NR	14 (2–27)–71% Stable
Falkenstein et al., 2018 [37]	NR	2.8 (95% CI = 2.0–3.7) ^p^	NR	NR	TRG0: 20%; TRG1-2: 40%; TRG3: 40%	20 [8–27]–NR
Larbre et al., 2018 [38]	NR	NR	NR	NR	NR	17 (5–39)–NR
Alyami et al., 2017 [39]	NR	NR	NR	NR	NR	19 (1–39) _1_–16 (1–39) _2_–15 (2–31) _3_
Farinha et al., 2017 [40]	NR	NR	NR	NR	NR	10 (5–17)–NR
Hilal et al., 2017 [41]	2.4 (0.3–21.7)	NR	NR	NR	NR	18.9 ± 11.2–NR
Khosrawipour et al., 2017 [42]	10.0	8.5 (95% CI = 8.5–11.9) ^p^	NR	NR	TRG1: 15%; TRG3: 25%; TRG4: 10%	26.6 (1–39)–NR
Demtroeder et al., 2016 [43]	22.0 ± 4	15.7 ^p^	NR	NR	TRG0: 11.7%; TRG1-2: 6%; TRG3: 24%; TRG4: 41.2%	16 ± 10–NR
Khomyakov et al., 2016 [44]	13.0	13.0 ^p^	NR	NR	1: 27%; 2: 33%; 3–4 40%	16 (6–34)–NR
Reymond et al., 2016 [25]	NR	11.7 (11.1–22) ^d^; 9.1 (8.6–18) ^p^	NR	PR 66%; SD 33%	1: 67%; 2: 33%	3 (3–17)–NR
Robella et al., 2016 [45]	NR	NR	NR	NR	NR	17 (12–21)–NR

_1, 2, 3_ measured at first, second and third PIPAC intervention respectively; * Radiological Response (RR) as reported by each study (e.g., SD = Stable Disease; PD = Progressive Disease; PR = Partial Response). ^p^ Survival calculated from the first PIPAC; ^d^ Survival calculated from diagnosis. Median overall months of survival reported with () for Standard Deviation and [] for Inter Quartile Range. Abbreviations: mo = months; OS = median Overall Survival; PFS = median Progression Free Survival; PRGS = peritoneal regression grading score; PCI = peritoneal cancer index; NR = not reported; TRG = tumor regression grade; CI = Confidence Interval. Pathologies: GC = Gastric Cancer; CRC = ColoRectal Cancer; PC = Pancreatic Cancer.

**Table 4 cancers-17-02580-t004:** Clinical outcomes stratified by primary tumor type.

Author & Year	Follow-Up (mo)	OS (mo)	PFS (mo)	RR *	PRGS	PCI
**Gastric Cancer (GC) **						
Sundar 2024 [28]	20.8	6.0 (2.4–18.4) ^p^	1.8 (1.6–6.0) ^p^	7%; SD 57%	1–2 in 66.7% _2_; 100% _3_	20 [12.3–30]; Δ−5/−7 _2,3_
Casella 2023 [30]	NR	19.6 (14–24) ^d^; 10.5 (7–13) ^p^	NR	NR	1 37.1%; 2 31.0%; 3 21.4% _2_;1 2.4%; 2 2.4%; 3 9.5% _3_	16 [8–26]
Alyami 2021 [33]	NR	19.1 ^p^	NR	NR	NR	17 (1–39)
Di Giorgio 2020 [35]	NR	12.3 (11.7–17.4) ^d^	NR	NR	CR 7.7%; PR 53.8%; SD 38.5%	20 ± 9.9; ↑ 76.9%; SD 7.7%; ↓ 15.4%
Khomyakov 2016 [44]	13.0	13.0 ^p^	NR	NR	1 27%; 2 33%; 3–4 40%	16 (6–34)
Kryh-Jensen 2023 [26] ^§^	NR	7.8 ^p^	NR	NR	↓ 61%	NR
**Colorectal Cancer (CRC)**						
Kyle 2023 [31]	11.6	11.6 (4.7–36) ^p^; 25.0 (7.2–42.2) ^d^	6.0 (0.5–25.1) ^p^	NR	NR	25 (20–32)
Demtrøder 2016 [43]	22.0 ± 4	15.7 ^p^	NR	NR	TRG0 11.7%; TRG1-2 6%; TRG3 24%; TRG4 41.2%	16 ± 10
Kryh-Jensen 2023 [26] ^§^	NR	16.0 ^p^	NR	NR	↓ 69%	NR
**Pancreatic Cancer (PC)**						
Kryh-Jensen 2023 [26] ^§^	NR	10.0 ^p^	NR	NR	↓ 59%	NR
Khosrawipour 2017 [42]	10.0	8.5 (95% CI 8.5–11.9) ^p^	NR	NR	TRG1 15%; TRG3 25%; TRG4 10%	26.6 (1–39)
**Diffuse Malignant Peritoneal Mesothelioma (DMPM)**						
Kepenekian 2021 [11]	29.6	12.0	12.0	15%	Pre 3.0 (2.2–4.0);Post 2.0 (2.0–3.0)	Pre 27 (20–34);Post 25 (20–39)
**Hepato-pancreato-biliary (HPB)** ^†^						
Falkenstein 2018 [37]	NR	2.8 (95% CI 2.0–3.7) ^p^	NR	NR	TRG0 20%; TRG1-2 40%; TRG3 40%	20 [8–27]

_2, 3_ measured at second and third PIPAC intervention respectively; ^p^ OS/PFS calculated from first PIPAC; ^d^ from diagnosis. ^§^ Study enrolled multiple tumor types but reported OS separately for the listed pathology; other outcomes were not stratified. ^†^ HPB series mainly comprised biliary-pancreatic primaries; data shown because the cohort contained a single pathology group. Abbreviations: mo = months; OS = Overall Survival; PFS = Progression-Free Survival; * RR = Radiological Response; PRGS = Peritoneal Regression Grading Score; PCI = Peritoneal Cancer Index; SD = Stable Disease; PD = Progressive Disease; CR = Complete Response; PR = Partial Response; TRG = Tumor Regression Grade; CI = Confidence Interval; NR = Not Reported.

**Table 5 cancers-17-02580-t005:** Summary of surgical (Clavien–Dindo) and medical (CTCAE) complications along with key other adverse events across the included studies.

Author & Year	Surgical (CD)	Medical (CTCAE)	Key Other AEs
Sundar et al., 2024 [28]	NR	G3: 50%; G4: 16.7%; G5: 11%	Pain; nausea/vomiting; weight loss; fatigue; etc.
van de Vlasakker et al., 2024 [29]	0%	NR	Mild GI symptoms
Casella et al., 2023 [30]	CD > 3a: 0%	G3/G4: 2%	NR
Kryh-Jensen et al., 2023 [26]	NR	NR	NR
Kyle et al., 2023 [31]	NR	G3: 20%	G1/G2: pain; nausea; constipation
Raoof et al., 2023 [32]	0%	G1: 58%; G2: 33%; G3: 17%	Pain; constipation; nausea/vomiting
Alyami et al., 2021 [33]	NR	G1–4: 6.1%; G3–4: 3.1%; G5: 4.7%	NR
Feldbrügge et al., 2021 [34]	CD ≤ 4: 11%; CD 3–4: 6%	NR	SSI
Kepenekian et al., 2021 [11]	NR	G1/G2: 10%; G3/G4: 3%	NR
Siebert et al., 2021 [27]	NR	>G3: 3.5%; mortality (30 d): 1.5%	Bowel obstruction; allergy
Di Giorgio et al., 2020 [35]	CD ≥ 3: 2%	G1–2: 22%; G3–4: 4%; G5 (30 d): 4%	NR
Gockel et al., 2020 [36]	0%	NR	6 non-access
Falkenstein et al., 2018 [37]	NR	G1: 41.4%; G2: 35.3%; G3–5: 0%	NR
Larbre et al., 2018 [38]	NR	NR	NR
Alyami et al., 2017 [39]	NR	G3–4: 9.7%; G5 (30d): 6.8%	Bowel obstruction; SSI
Farinha et al., 2017 [40]	NR	NR	Fatigue; GI symptoms
Hilal et al., 2017 [41]	NR	NR	NR
Khosrawipour et al., 2017 [42]	NR	G1: 34%; G2: 2.4%; G3–4: 0%; G5: 2.4%	NR
Demtröder et al., 2016 [43]	0%	G1: 71%; G3: 23%; G4–5: 0%	Nausea/vomiting; renal/liver toxicity G1
Khomyakov et al., 2016 [44]	NR	G2: 6.4%; G3: 3.2%; G4–5: 0%	NR
Reymond et al., 2016 [25]	NR	G1: 66%; G2: 33%	Pain
Robella et al., 2016 [45]	0%	G1: 46%; G2: 62%	Pain; nausea

Abbreviations: CD = Clavien-Dindo classification: CDX = classification; CTCAE = Common Terminology Criteria
for Adverse Events: GX = Grade; NR = not reported; GI = Gastro Intestinal; SSI = Surgical Site Infection.

**Table 7 cancers-17-02580-t007:** Evaluation of items I1–I8 and overall score for included studies.

Author & Year	I1	I2	I3	I4	I5	I6	I7	I8	Overall
Sundar et al., 2024 [28]	2	1	2	2	1	2	2	1	13
van de Vlasakker et al., 2024 [29]	2	2	2	2	1	2	0	1	12
Casella et al., 2023 [30]	2	2	2	2	1	2	2	0	13
Kryh-Jensen et al., 2023 [26]	2	2	1	2	1	2	2	0	12
Kyle et al., 2023 [31]	2	2	2	2	1	2	2	1	14
Raoof et al., 2023 [32]	2	1	2	2	1	2	2	1	13
Alyami et al., 2021 [33]	2	2	2	2	1	2	2	0	13
Feldbrügge et al., 2021 [34]	2	2	1	1	1	2	2	0	11
Kepenekian et al., 2021 [11]	2	2	2	2	1	2	2	0	13
Di Giorgio et al., 2020 [35]	2	2	1	1	1	2	2	0	11
Gockel et al., 2020 [36]	2	2	2	2	1	2	2	0	13
Falkenstein et al., 2018 [37]	2	2	2	2	1	2	2	0	13
Larbre et al., 2018 [38]	2	2	2	2	1	2	2	0	13
Alyami et al., 2017 [39]	2	2	2	2	1	2	2	0	13
Farinha et al., 2017 [40]	2	2	2	2	1	2	2	0	13
Hilal et al., 2017 [41]	2	2	2	2	1	2	0	0	11
Khosrawipour et al., 2017 [42]	2	2	2	2	1	2	2	0	13
Demtröder et al., 2016 [43]	2	2	2	2	1	2	2	0	13
Khomyakov et al., 2016 [44]	2	2	2	2	1	2	0	0	11
Reymond et al., 2016 [25]	2	2	2	2	1	2	2	0	13
Robella et al., 2016 [45]	2	2	2	2	1	2	2	0	13

I1: Clearly stated aim; I2: Inclusion of consecutive patients; I3: Prospective data collection; I4: Appropriate endpoints; I5: Blinding of outcome assessment; I6: Unbiased endpoint measurement; I7: <5 % loss to follow-up; I8: Prospective sample size calculation.

**Table 8 cancers-17-02580-t008:** Extended evaluation of bias domains for the additional study.

Author & Year	I1	I2	I3	I4	I5	I6	I7	I8	I9	I10	I11	I12	Overall
Siebert et al., 2021 [27]	2	2	1	2	1	0	0	0	2	2	1	1	14

I1: Clearly stated aim; I2: Inclusion of consecutive patients; I3: Prospective data collection; I4: Appropriate endpoints; I5: Blinding of outcome assessment; I6: Unbiased endpoint measurement; I7: <5% loss to follow-up; I8: Prospective sample size calculation; I9: Adequate control group; I10: Contemporary groups; I11: Baseline equivalence of groups; I12: Adequate statistical analysis.

## Data Availability

The datasets used and/or analyzed during the current study are available from the corresponding author on reasonable request.

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
