# Peer review of "Advances in Bidirectional Therapy for Peritoneal Metastases: A Systematic Review of Pressurized Intraperitoneal Aerosol Chemotherapy (PIPAC) Combined with Systemic Chemotherapy"

_cancers, 2025, doi:10.3390/cancers17152580_

Round 1

Reviewer 1 Report

Comments and Suggestions for Authors

Intraperitoneal Aerosol Chemotherapy (PIPAC) for peritoneal malignancies, analyzing 22 recent studies with a focus on efficacy, toxicity, and study quality. While the effort to synthesize the current literature is commendable, the heterogeneity in study designs—particularly in terms of primary tumor types, chemotherapeutic regimens, and treatment intervals—limits the ability to draw definitive conclusions. I offer the following comments and recommendations to enhance the clarity and utility of this review:

  1. Stratification by Primary Tumor Type:
    Survival and tumor response outcomes vary substantially by the origin of the primary tumor. Specifically, outcomes tend to be more favorable in ovarian carcinoma, less so in colorectal cancer, and poorest in gastric and pancreatic carcinomas. To improve interpretability, I recommend that the authors stratify and present clinical outcomes separately by primary tumor type. This stratification should be reflected in a revised version of Table 3.
  2. Toxicity Data and Methodological Variables:
    Reported toxicity appears to depend heavily on treatment methodology, including the choice and dose of intraperitoneal agents, frequency of PIPAC administration, and concurrent use of systemic therapy. Table 4 should be expanded to include these critical variables, which would allow readers to better contextualize toxicity outcomes.
  3. Completeness of Ongoing Clinical Trials (Table 5):
    Table 5 contains valuable information on ongoing trials; however, it would benefit from additional key data. I suggest incorporating the following elements for each trial: targeted disease, eligibility criteria, PIPAC drug dosage, and treatment interval. This information is essential for evaluating trial design and relevance to clinical practice.

Overall, with these revisions, the manuscript would provide a more nuanced and clinically meaningful overview of the current PIPAC literature.

Author Response

Stratification by Primary Tumor Type:
Survival and tumor response outcomes vary substantially by the origin of the primary tumor. Specifically, outcomes tend to be more favorable in ovarian carcinoma, less so in colorectal cancer, and poorest in gastric and pancreatic carcinomas. To improve interpretability, I recommend that the authors stratify and present clinical outcomes separately by primary tumor type. This stratification should be reflected in a revised version of Table 3.

We thank the reviewer for this important suggestion. As recommended, we have stratified and presented clinical outcomes by primary tumor type where such data were explicitly available. This stratification is already reflected in Table 3 of the main manuscript. However, given that several included studies enrolled patients with multiple tumor types but reported outcomes without subgroup separation, a complete stratification was not feasible for all cohorts. To enhance transparency and completeness, we have provided a detailed breakdown by primary tumor type—wherever possible—in the table 4.

Toxicity Data and Methodological Variables:
Reported toxicity appears to depend heavily on treatment methodology, including the choice and dose of intraperitoneal agents, frequency of PIPAC administration, and concurrent use of systemic therapy. Table 4 should be expanded to include these critical variables, which would allow readers to better contextualize toxicity outcomes.

We thank the reviewer for this thoughtful suggestion. However, we opted not to incorporate all treatment-related variables into Table 4 (now table 5) , as doing so would result in an overly complex and potentially confusing table, given the heterogeneity of protocols and reporting across studies. The type of systemic chemotherapy, intraperitoneal agents used in PIPAC, and their respective dosages are already detailed in Table 2. Additionally, in the discussion section, we have commented on the likely source of adverse events—whether attributable to surgery, systemic therapy, or PIPAC—whenever this was clearly specified by the original study authors. We believe this approach maintains clarity while still providing readers with the necessary contextual information.

Completeness of Ongoing Clinical Trials (Table 5):
Table 5 contains valuable information on ongoing trials; however, it would benefit from additional key data. I suggest incorporating the following elements for each trial: targeted disease, eligibility criteria, PIPAC drug dosage, and treatment interval. This information is essential for evaluating trial design and relevance to clinical practice

We thank the reviewer for the constructive feedback. In response, we have revised the table  to include information on tumor type and eligibility criteria for each ongoing trial. However, details regarding PIPAC drug dosages and treatment intervals were not consistently reported in the available protocols or trial registries, and therefore could not be reliably included. We believe the updated table provides a more informative and clinically relevant overview while maintaining clarity and consistency.

Reviewer 2 Report

Comments and Suggestions for Authors

This systematic review analyzed 22 studies on PIPAC combined with systemic chemotherapy for non-resectable peritoneal metastasis. Results indicate potential benefits in symptom control and tumor response; however, data on survival and toxicity are inconsistent. More standardized, high-quality studies are needed to confirm its clinical value. Authors are advised to revise their manuscript based on the following technical comments -

  1. The title is nearly identical to that of a previously published review by Ploug et al. (2019). Rephrase it to reflect its updated scope better and clearly distinguish it from prior work.
  2. The abstract can be more precise in quantifying results and clearer in phrasing limitations to enhance clarity and impact. The abstract should indicate the objective of this study clearly.
  3. At the beginning of the Introduction (1.1), authors are advised to discuss the limitations of conventional chemotherapy in cancer treatment to introduce PIPAC. Authors are also advised to cite the following articles - https://doi.org/10.3390/genes14071370
  4. The latter part of the Introduction is clear and provides a solid rationale for the review. It could more clearly highlight what makes this review different from the review published by Ploug et al. (2019). This will be considered the novelty of this study.
  5. The methodology is clear and follows PRISMA guidelines, with a PROSPERO registration and clearly defined inclusion criteria. The authors need to explain how the data were combined, as a meta-analysis was not performed.
  6. Results: The studies included in the review differ significantly from one another, making it unclear how the authors directly compared them.
  7. There is an inconsistency in how tumor response is reported across the included studies, which makes direct comparison difficult.
  8. The authors are advised to describe how missing data was handled.
  9. Discussion: Many patients still had disease progression or died after PIPAC, so the authors should explain if its main benefit is symptom relief, better quality of life, or making surgery possible, not just longer survival.
  10. The discussion section should also discuss how future research can improve aspects such as reporting and selecting the right patients.
  11. The sample size is slightly smaller. Is it statistically significant? As a reviewer, I want to know the authors' comments, and the authors can include a couple of lines to establish their opinion in the manuscript at the end of the discussion and conclusion.
  12. It is suggested to make some language corrections and edit grammatical errors in certain places.
Comments on the Quality of English Language

A few language errors were observed. Authors should check their English language and grammar.

Author Response

- The title is nearly identical to that of a previously published review by Ploug et al. (2019). Rephrase it to reflect its updated scope better and clearly distinguish it from prior work.

We thank the reviewer for this helpful observation. In response, we have revised the title to better reflect the scope and novelty of our review. The new title—“Advances in Bidirectional Therapy for Peritoneal Metastases: A Systematic Review of PIPAC Combined with Systemic Chemotherapy”—emphasizes both the updated evidence base and the integration of treatment modalities, clearly distinguishing this work from previous reviews.

- The abstract can be more precise in quantifying results and clearer in phrasing limitations to enhance clarity and impact. The abstract should indicate the objective of this study clearly.

We thank the reviewer for this valuable suggestion. In response, we have revised the abstract to state the objective of the review more clearly and concisely.

- At the beginning of the Introduction (1.1), authors are advised to discuss the limitations of conventional chemotherapy in cancer treatment to introduce PIPAC. Authors are also advised to cite the following articles - https://doi.org/10.3390/genes14071370

We thank the reviewer for the suggestion. As recommended, we have emphasized the limitations of systemic chemotherapy—particularly in the context of peritoneal metastases—early in the Introduction to better justify the rationale for PIPAC (lines 46-56) . Regarding the proposed citation, we respectfully chose not to include it, as the article focuses primarily on molecular profiling and does not directly address intraperitoneal therapies or the pharmacologic limitations of conventional chemotherapy in peritoneal disease.

- The latter part of the Introduction is clear and provides a solid rationale for the review. It could more clearly highlight what makes this review different from the review published by Ploug et al. (2019). This will be considered the novelty of this study.

We thank the reviewer for this useful suggestion. We have clarified how our review differs from the earlier work by Ploug et al. (2019) in lines 71–75 of the revised manuscript.

- The methodology is clear and follows PRISMA guidelines, with a PROSPERO registration and clearly defined inclusion criteria. The authors need to explain how the data were combined, as a meta-analysis was not performed.

We thank the reviewer for this observation. As clarified in lines 154–163 of the manuscript, we conducted a narrative synthesis of the available data, as the heterogeneity in study design, patient populations, treatment protocols, and outcome reporting precluded formal meta-analysis. This approach is consistent with PRISMA guidelines and was chosen to allow a comprehensive and clinically meaningful summary of findings across a broad range of studies.

- Results: The studies included in the review differ significantly from one another, making it unclear how the authors directly compared them.

We thank the reviewer for this important point. As noted in lines 139–142 of the manuscript, outcomes were reported using different summary statistics (e.g., median [range], mean [SD], median [IQR]). To preserve the integrity of the original data and maximize inclusion, we retained each study’s reported format without performing data transformation. No direct comparisons between studies were made; instead, we opted for a narrative synthesis to summarize results descriptively. Any discrepancies in interpretation were reconciled through discussion among the authors.

- There is an inconsistency in how tumor response is reported across the included studies, which makes direct comparison difficult.

We fully agree with the reviewer’s observation. As noted in the results session, the included studies used varying response assessment methods—such as PRGS, radiologic criteria, or cytology—often with non-uniform timing or incomplete reporting, which limits direct comparability. We also addressed this issue in the disccusion, recognizing small sample sizes and outcome heterogeneity as key challenges. To address this, we explicitly call for standardized endpoints in future research  including consistent histologic response criteria, harmonized definitions of complications, and validated quality-of-life instruments. We believe this will be critical for improving comparability and interpretability across future trials

- The authors are advised to describe how missing data was handled.

We thank the reviewer for raising this point. As specified in lines 139–142, we retained each study’s original summary statistics (e.g., median [range], mean [SD], median [IQR]) to ensure maximal inclusion and avoid data transformation that could compromise accuracy. In cases where outcome data were incomplete or inconsistently reported, we did not impute or estimate missing values. Instead, any such limitations were clearly acknowledged in the tables or narrative synthesis. Discrepancies in interpretation were resolved through discussion among the authors.

 - Discussion: Many patients still had disease progression or died after PIPAC, so the authors should explain if its main benefit is symptom relief, better quality of life, or making surgery possible, not just longer survival.

We appreciate this important observation. As noted in lines 495–498, quality of life (QoL) was assessed in six studies using validated instruments such as the EORTC QLQ-C30 and SF-36. These studies reported stable or improved QoL scores during PIPAC treatment, which supports its role in symptom control and functional maintenance. While extended survival remains an important endpoint, we agree that palliation, preservation of quality of life, and in selected cases conversion to surgery represent meaningful clinical goals of bidirectional therapy. We have added a clarifying statement in the Discussion to reflect this broader therapeutic perspective.

- The discussion section should also discuss how future research can improve aspects such as reporting and selecting the right patients.

We thank the reviewer for this valuable suggestion. We agree that both standardized reporting and appropriate patient selection are critical areas for future development. These points are explicitly addressed in the Discussion, particularly in lines 519–529 and 540–541

- The sample size is slightly smaller. Is it statistically significant? As a reviewer, I want to know the authors' comments, and the authors can include a couple of lines to establish their opinion in the manuscript at the end of the discussion and conclusion.

We thank the reviewer for the comment. While our review includes a relatively large pooled cohort (over 1,000 patients across 22 studies), we recognize that the majority of the individual studies are retrospective and non-comparative, and therefore not powered to demonstrate statistical significance. As this is a systematic review without pooled meta-analytic analysis, statistical significance is not applicable to the aggregate sample. Nonetheless, the number of patients included represents a substantial body of clinical experience with bidirectional therapy.

Reviewer 3 Report

Comments and Suggestions for Authors

Robella and team presented an interesting systematic review manuscript titled “Bidirectional Treatment of Peritoneal Metastasis (PM) with Pressurized Intraperitoneal Aerosol Chemotherapy (PIPAC) and Systemic Chemotherapy: a systematic review”. The authors aimed to study the role of dual-directional therapy framework in improving quality of drugs administrations in the management of cancer. The authors covered data from 2011 to 2024 pertaining to 22 studies covering both retrospective, and prospective studies. The introduction section encompassed the basic information with literature support. Followed by introduction, the author detailed methodology followed by results including an extended evaluation. The manuscript also included Tables, Figure and presented clearly. The discussion part also included current limitation and future perspectives. The authors concluded that bidirectional therapy is a feasible and safe approach with promising signals of efficacy in selected patients with PM. This study was registered with PROSPERO in accordance with PRISMA-P guidelines which is quite appreciable. Scientifically, this study is perfect, suitable for publication. However, a revision is required before the publication.

  1. The peritoneal metastases mortality rate is to be mentioned in the introduction section.
  2. Wherever applicable, authors need to cite recent literature reports and increase the number of references.
  3. Before references, other missing sections needed to be included, such as an ethical statement, acknowledgements, and authors' contributions. The authors need to follow the guidelines of the journal, accordingly.

Author Response

- The peritoneal metastases mortality rate is to be mentioned in the introduction section.

We appreciate the reviewer’s suggestion. The mortality and survival data related to peritoneal metastases have been explicitly included in the Introduction, lines 49–55, with specific reference to gastric, colorectal, pancreatic, and ovarian cancers. These data underscore the clinical urgency and justify the rationale for exploring bidirectional therapeutic approaches such as PIPAC.

 - Wherever applicable, authors need to cite recent literature reports and increase the number of references.

We appreciate the reviewer’s recommendation. In response, we have revised and expanded the reference list to include several recent publications on PIPAC, ensuring that the manuscript reflects current evidence and aligns with the latest developments in the field.

- Before references, other missing sections needed to be included, such as an ethical statement, acknowledgements, and authors' contributions. The authors need to follow the guidelines of the journal, accordingly.

We thank the reviewer for pointing this out. In accordance with the journal’s guidelines, we have now included the required sections: Ethical Statement, Acknowledgements, and Authors’ Contributions prior to the References. These additions ensure full compliance with the editorial standards of the journal.

Round 2

Reviewer 1 Report

Comments and Suggestions for Authors

All my points are appropriately revised, and could be accepted as the present form.

Reviewer 2 Report

Comments and Suggestions for Authors

May be accepted for publication.